# Integration of Eukaryotic Energy Metabolism: The Intramitochondrial and Cytosolic Energy States ([ATP]_f_/[ADP]_f_[Pi])

**DOI:** 10.3390/ijms23105550

**Published:** 2022-05-16

**Authors:** David F. Wilson, Franz M. Matschinsky

**Affiliations:** Department of Biochemistry and Biophysics, Perelman School of Medicine, University of Pennsylvania, 422 Curie Blvd, Philadelphia, PA 19104, USA; matsch@pennmedicine.upenn.edu

**Keywords:** mitochondrial matrix, cytoplasm, metabolic integration, energy metabolism

## Abstract

Maintaining a robust, stable source of energy for doing chemical and physical work is essential to all living organisms. In eukaryotes, metabolic energy (ATP) production and consumption occurs in two separate compartments, the mitochondrial matrix and the cytosol. As a result, understanding eukaryotic metabolism requires knowledge of energy metabolism in each compartment and how metabolism in the two compartments is coordinated. Central to energy metabolism is the adenylate energy state ([ATP]/[ADP][Pi]). ATP is synthesized by oxidative phosphorylation (mitochondrial matrix) and glycolysis (cytosol) and each compartment provides the energy to do physical work and to drive energetically unfavorable chemical syntheses. The energy state in the cytoplasmic compartment has been established by analysis of near equilibrium metabolic reactions localized in that compartment. In the present paper, analysis is presented for energy-dependent reactions localized in the mitochondrial matrix using data obtained from both isolated mitochondria and intact tissues. It is concluded that the energy state ([ATP]_f_/[ADP]_f_[Pi]) in the mitochondrial matrix, calculated from the free (unbound) concentrations, is not different from the energy state in the cytoplasm. Corollaries are: (1) ADP in both the cytosol and matrix is selectively bound and the free concentrations are much lower than the total measured concentrations; and (2) under physiological conditions, the adenylate energy states in the mitochondrial matrix and cytoplasm are not substantially different.

## 1. Introduction

### Intramitochondrial Metabolism in Eukaryotes

It is generally believed that mitochondria originated when archaea, motile nucleated cells with Embden-Meyerhof glycolysis as their energy source, engulfed and symbiotically assimilated an α-proteobacterial ancestor of modern *paracoccus denitrificans* [1,2,3]. The two organisms had many metabolic processes in common, although utilizing different primary metabolic energy sources, glycolysis and oxidative phosphorylation. During assimilation, much of the genetic information from the endocytosed α-proteobacterium was transferred to the host, but some genetic code and extensive metabolism remained in what became the mitochondrial matrix. To understand cellular metabolism as a whole, it is important to understand energy metabolism in each compartment (mitochondrial matrix and cytoplasm) and how that metabolism is coordinated. Central to energy metabolism is the adenylate energy state ([ATP]/[ADP][Pi]). Hydrolysis of the terminal phosphate of ATP synthesized by oxidative phosphorylation (mitochondrial matrix) and glycolysis (cytosol) provides the energy to do physical work and drive energetically unfavorable chemical syntheses. Cytoplasmic metabolism has been more readily experimentally accessed than metabolism localized in the mitochondrial matrix, but many of the reactions/metabolic pathways involve chemically similar reactions. These include synthesis of macromolecules such as proteins [4], DNA [5], and RNA [6], as well as a wide range of other energy-coupled reactions. The latter include the nucleoside mono- and di-phosphokinases, carbamoyl-phosphate synthetase, acyl carnitine transferase, phosphoenolpyruvate carboxykinase, propionyl-CoA carboxylase, succinate thiokinase, butyryl-CoA ligase, pyruvate carboxylase, etc. In addition to endothermic synthetic reactions in which energy input, through coupling to hydrolysis of a phosphate bond of a nucleoside triphosphate, is required, there are enzymes catalyzing phosphate exchange among the nucleoside mono-, di-, and tri- phosphates and enzymes regulated by those nucleotides. The metabolic energy state based on the free (unbound) concentrations of ATP, ADP, and Pi ([ATP]_f_/[ADP]_f_[Pi]) has been established for the cytoplasm [7], but this in the mitochondrial matrix remains controversial. We have analyzed available data on metabolic reactions that occur in, and can be used to infer the energy state ([ATP]_f_/[ADP]_f_[Pi]) of, the mitochondrial matrix. This will allow comparison of the energy state in the matrix with that in the cytoplasm and thereby provide a better understanding of how metabolism in the cytoplasm and mitochondrial matrix is coordinated.

Our focus is on the thermodynamics of the hydrolysis of the terminal phosphate of ATP and its coupled reactions under in vivo conditions. Knowledge of these thermodynamic parameters is essential to understanding energy metabolism in the matrix and, by comparison with that in the cytoplasm, to an integrated whole. Since thermodynamic values (equilibrium constants, free energy, etc.) are state functions, they are not dependent on reaction mechanism or rates at which the reaction(s) occur. Thermodynamic measurements cannot be used to determine the mechanism of a reaction, but they place essential constraints on the properties of proposed mechanisms. As a result, the discussion of proposed mechanisms of adenine nucleotide and Pi transport across the mitochondrial membrane are minimal, limited to illustrations of the necessity for applying thermodynamic constraints to proposed mechanisms. Any mechanism proposed for the transport of the nucleotides and Pi between the compartments must predict outcomes (states) consistent with the relationship of the energy states in the two compartments. In addition, the ratio of the proposed forward and reverse reaction rates must be consistent with the measured displacement from equilibrium (degree of reversibility).

## 2. Results and Discussion

### 2.1. The Intramitochondrial [ATP]/[ADP][Pi]: A. In Vitro Measurement of Nucleotide Concentrations

Many research groups have reported measurements of the intramitochondrial nucleotide concentrations for suspensions of isolated mitochondria. The most widely used experimental method involves incubation of mitochondrial suspensions under defined conditions followed by placing aliquots in centrifuge tubes with a layer of quenching solution covered by a layer of silicone oil. Centrifugation then causes the mitochondria to rapidly pass through the layer of oil into a quenching solution, typically a perchloric acid solution. The nucleotides carried through the oil and into the quenching layer with the mitochondria, corrected for suspending medium carried through the oil, are then considered to represent the content of the mitochondrial matrix [8,9,10,11,12,13,14]. Measurements for suspensions of isolated mitochondria provided with oxidizable substrate, oxygen, and Pi, then allowed to maximally phosphorylate added ADP (State 4), have been reported by several laboratories. Under these conditions it is generally reported that the ratio of measured (total) concentrations ([ATP]_T_/[ADP]_T_) in the mitochondria is 2–8 whereas that in the suspending medium is 100–600 [9,10,11,12]. Slater et al. [9] calculated the apparent difference in free energy of hydrolysis of ATP, assuming the Pi, ADP, and ATP in the matrix were all free in the solution. After accounting for differences in [Mg^2+^] and pH between the suspending medium and matrix, the authors calculated a difference of −4.5 kcal/mole (−18.8 kJ/mole). Others [10,15] have reported lower values. Wilson and coworkers [16] evaluated binding of ATP and ADP to mitochondrial components in detergent permeabilized rat liver mitochondria (final detergent concentrations 1% Triton, 1% sodium deoxycholate) after adding sufficient purified nucleotide diphosphate kinase and creatine kinase to establish rapid equilibration of their respective reactions. High-pressure liquid chromatography was used to measure nucleotide concentrations as a function of the creatine phosphate/creatine ([CrP]/[Cr]) ratio. The relationships of [ATP]/[ADP], [UTP]/[UDP], and [GTP]/[GDP] were consistent with ADP being selectively bound to sites present at about 0.4–1.0 nmol/mg protein and a Kd < 10 µM. ATP was bound to a similar number of sites, but with about a 100-fold lower affinity. This is a sufficient number of sites to bind most of the intramitochondrial ADP. Adenine nucleotide translocase, for example, is present at about 0.4 n mole/mg and has been reported to bind ADP with a Kd ≈ l–2 µM and ATP with a Kd ≥ 50 µM [17,18]. It was concluded that selective binding of intramitochondrial ADP was likely responsible for the observed difference between the mitochondrial [ATP]_T_/[ADP]_T_ and extramitochondrial [ATP]_f_/[ADP]_f_. In detergent permeabilized mitochondria, however, ADP binding sites exposed to the cytoplasm could not be distinguished from those in the mitochondrial matrix, and a definitive conclusion was not possible.

### 2.2. The Intramitochondrial [ATP]/[ADP][Pi]: B. the Effect of pH and Membrane Potential on Transport of ATP and ADP across the Inner Membrane

The inner mitochondrial membrane is semipermeable and transport of ions and metabolites across the membrane may be coupled to other gradients across the membrane (pH, ion, or electrical). For net transport of uncharged weak acids across the membrane, the limiting distribution between the extra- and intra-mitochondrial spaces is a function of the difference in pH. The pH difference and its effect on adenine nucleotide transport, which occurs by stoichiometric ATP–ADP exchange (no net change in total intramitochondrial adenine nucleotide concentration), has been measured [9,11,12,19,20]. There is general agreement that the effect of pH on ATP–ADP exchange across the membrane is limited to the difference in protonation of the nucleotides in the intra- and extra-mitochondrial spaces.

Similar agreement on the dependence on the membrane potential has not been achieved, as exemplified by the reports of Slater et al. [9] and Klingenberg and Rottenberg [19]. Slater and coworkers [9] reported that although in in vitro experiments the extramitochondrial [ATP]_f_/[ADP]_f_ greatly exceeded the intramitochondrial [ATP]_T_/[ADP]_T_ (see above), there was little difference whether the mitochondria were in State 4 (high membrane potential) or uncoupled (low membrane potential). This contrasted with the report of Klingenberg and Rottenberg [19]. The latter measured the distribution of isotopically labeled nucleotides, but did not report the concentrations of ATP or ADP, only logarithms of the calculated [ATP]/[ADP]. They reported that the logarithm of the intramitochondrial [ATP]/[ADP], when measured for mitochondria with oxidative phosphorylation blocked and at constant extramitochondrial [ATP]/[ADP], was a linear function of the membrane potential with an average slope of 0.85. Notably, the graphs presented by Klingenberg and Rottenberg [19] show intramitochondrial [ATP]/[ADP] decreasing about 100 fold relative to the extramitochondrial [ATP]/[ADP] as the membrane potential was increased from near zero to −160 mV. In the literature, there are several reports of smaller differences between the measured intramitochondrial [ATP]_T_/[ADP]_T_ and extramitochondrial [ATP]/[ADP]. Tager et al. [15] reported differences of 16 and 7.2 fold in rat liver mitochondria when the respiration was stimulated to about 44% of maximum by citrulline synthesis or glucose plus hexokinase, respectively. A general observation is that the intramitochondrial [ATP]_T_/[ADP]_T_ remains relatively constant during large changes in mitochondrial respiratory rate and other activities associated with energy metabolism including conditions giving rise to large changes in cytoplasmic [ATP]/[ADP].

### 2.3. In Vivo Measurements of Cytoplasmic [ATP]/[ADP]: The Difference between Cytoplasmic [ATP]_T_/[ADP]_T_ and [ATP]_f_/[ADP]_f_


The [ATP]_T_/[ADP]_T_ reported for the matrix of isolated mitochondria is between one and eight, similar to that measured for the total content in intact cells and tissues [7,15,21,22,23,24,25]. It was recognized early that, for the cytoplasm of muscle cells, this low ratio was not compatible with the observed [CrP]/[Cr]. Creatine phosphate synthesis/consumption occurs in the cytoplasm and is a “dead end” reaction. The reaction is catalyzed by creatine kinase and its equilibrium constant [26] is:CrP + ADP = Cr + ATP   Keq ≈ 150

The [CrP]/[Cr] observed in resting muscle tissues is about two [7,25], requiring the cytoplasmic [ATP]_f_/[ADP]_f_ to be approximately 300. Veech and coworkers [7] used the mass action ratios for this and other near equilibrium reactions in the cytoplasm to calculate the cytoplasmic [ATP]_f_/[ADP]_f_ for red cells, liver, brain, and heart tissue. It was concluded that in all of these tissues, only about 5% of the ADP is free in solution. Therefore, in the cytoplasm in vivo, [ATP]_f_/[ADP]_f_ is about 20 times higher than [ATP]_T_/[ADP]_T_. It is now generally accepted that the cytoplasmic [ATP]_f_/[ADP]_f_ is much higher than [ATP]_T_/[ADP]_T_, and that in tissues with creatine kinase activity, calculation from [CrP]/[Cr] provides a good measure of the cytoplasmic [ATP]_f_/[ADP]_f_. Inorganic phosphate, which is at higher concentrations than ADP, may also be significantly bound. Iles et al. [27] used MRI to estimate [Pi]_f_ in perfused rat liver and concluded that approximately 1/3 of the total is free in solution. The fraction of [Pi]_T_ that is free is quite variable among tissues and metabolic conditions, such as during the large changes in [Pi] that occur in muscle. In general, the correction for Pi binding is much smaller than for ADP and proportionally less important in calculating the energy state. Thus, there is agreement that the cytoplasmic energy state in eukaryotes, based on free concentrations (activities), of ATP, ADP, and Pi under physiological conditions is 50 to 100 times greater than that calculated from the total (measured) concentrations.

In much of the literature, the intramitochondrial [ATP]_T_/[ADP]_T_ is compared to the extramitochondrial [ATP]_f_/[ADP]_f_ and the difference ascribed to the mitochondrial membrane potential. This is a logical error often referred to as a comparison of apples to oranges. When the cytoplasmic [ATP]_T_/[ADP]_T_ is compared to that in the matrix, the values are not significantly different. In order to compare [ATP]_f_/[ADP]_f_ in the cytoplasm to that in the mitochondrial matrix, it is essential that the comparison is logically correct, i.e., that apples are compared to apples. To do that, it is necessary to establish the extent to which the nucleotides and Pi in the mitochondrial matrix are bound, as Veech and coworkers [7] did for the cytoplasm.

### 2.4. The Intramitochondrial Energy State ([ATP]/[ADP][Pi]) Inferred by Metabolic Reactions Localized to the Matrix: A. Nucleoside Kinases

Nucleoside kinases, of which adenylate kinase is the most widely known, catalyze transfer of the terminal phosphate of nucleoside triphosphates to nucleoside monophosphates producing two nucleoside diphosphates. For adenylate kinase, the reaction is:AMP + ATP = 2 ADP   Keq ≈ 1.0 

This activity is essential to energy metabolism, re-phosphorylating monophosphates formed in biosynthetic reactions (fatty acid activation, etc.), as well as helping to maintain the nucleotide pools through de novo synthesis and nucleoside salvage. Adenylate kinase activity localized in the mitochondrial matrix has been reported [14,28]. The adenylate kinase reaction has been established as being near equilibrium in the cytoplasm of intact cells and is often used to calculate the concentration of free AMP ([AMP]_f_). Less is known about adenylate kinase activity in the mitochondrial matrix. Roberts and coworkers [14] carried out a comprehensive study of nucleotide levels and nucleoside phosphorylation in suspensions of potato mitochondria and their perchloric acid extracts using NMR. They reported that the capacity for adenylate kinase and nucleoside diphosphate kinase to equilibrate the nucleoside mono-, di-, and triphosphates pools inside the mitochondria exceeded the maximal rate of ATP synthesis. Potato mitochondria could phosphorylate added AMP, UDP, IDP, and GDP to the corresponding triphosphates at rates similar to that for ADP, consistent with the matrix having sufficient AK and NDK activity to equilibrate the intramitochondrial nucleotide pools.

Krebs and Wiggins [29] determined the rate of AMP production in the mitochondrial matrix of intact rat hepatocytes by butyryl-CoA ligase (AMP-forming, EC 6.2.1.2) by measuring the rate of ketone formation. For this reaction to proceed, AMP needs to be re-phosphorylated by GTP-AMP transphosphorylase:GTP+AMP=GDP+ADP Keq ≈1.0

The authors concluded that GTP levels were not limiting and that in liver mitochondria in vivo the rate of GTP regeneration from ATP was substantially higher than the maximal rates of AMP formation, i.e., most of the GTP consumed by butyryl-CoA ligase was synthesized from GDP by nucleoside diphosphokinase (EC 2.7.4.6):ATP+GDP=ADP+GTP Keq ≈1.0

Garber and Hanson [30] measured the nucleoside diphosphate kinase activity in liver mitochondria from mitochondria from rabbits, guinea pigs, and rats, The activity in rabbit liver mitochondria was lowest (1.35 units/g liver), but that activity was sufficient to support about 50% of the GTP used to synthesize phosphoenolpyruvate for gluconeogenesis. The activities in guinea pig and rat liver mitochondria were substantially higher, 5.38 and 9.85 units/g liver, respectively. The general conclusion was that the rate of phosphate transfer among nucleotides within the mitochondrial matrix in liver is rapid, exceeding the rates of consumption of the individual nucleotides.

### 2.5. The Intramitochondrial Energy State ([ATP]/[ADP][Pi]) Inferred by Metabolic Reactions Localized to the Matrix: B. Phosphoenoylpyruvate Carboxykinase

PEP carboxykinase (EC 4.1.1.32) plays an important role in liver gluconeogenesis in animals and catalyzes the reaction:PEP + CO_2_ + MgGDP = Oxaloacetate + MgGTP   Keq ≈ 0.2 M^−1^

The enzyme is localized in the cytoplasm of rats and mice, but is in both the cytoplasm and mitochondria in rabbits, humans, and guinea pigs [11,12,30,31]. In birds, the activity is almost exclusively in the mitochondria [11,12]. Wilson et al. [11] measured the forward and reverse reactions of PEP carboxykinase in suspensions of mitochondria isolated from the livers of guinea pigs, pigeons, and chickens under conditions where oxidative phosphorylation served as the source of ATP, adjusting the initial conditions for net synthesis of OAA or net production of PEP. The incubations were then continued until product accumulation/substrate depletion resulted in reversal of the direction of the net flux. The intramitochondrial metabolite concentrations at the point where the net flux reversed were used to calculate the mass action ratio. At the reversal point, the calculated mass action ratio was, within experimental error, not different from the equilibrium constant. This is consistent with the reaction being near equilibrium. It was concluded that the intramitochondrial [ATP]_f_/[ADP]_f_ was not significantly different from the extramitochondrial [ATP]/[ADP]. Erecińska and Wilson [12] then made measurements under conditions for which the GTP was entirely provided by extramitochondrial ATP in combination with nucleoside diphosphate kinase activity in the matrix. Rotenone, oligomycin, and uncoupler (carbonylcyanide p-trifluoromethoxyphenylhydrazone} were added to prevent production of ATP by oxidative phosphorylation. The extramitochondrial ATP/ADP was controlled by adding creatine kinase and different [CrP]/[Cr] ratios to the incubation medium. At the point where the PEP carboxykinase reaction changed from net synthesis of oxaloacetate to net consumption of oxaloacetate, the calculated intramitochondrial [GTP]_f_/[GDP]_f_ was not significantly different from the extramitochondrial [ATP]/[ADP]. The authors concluded that the matrix energy state was not different from that in the suspending medium, and that ATP/ADP exchange across the mitochondrial inner membrane occurs by exchange diffusion and is not energy dependent.

### 2.6. The Intramitochondrial Energy State ([ATP]/[ADP][Pi]) Inferred by Metabolic Reactions Localized to the Matrix: C. Succinate Thiokinase (Succinate CoA Ligase)

Succinate thiokinase is localized exclusively in the mitochondrial matrix and is responsible for substrate level phosphorylation in the citric acid cycle:succ-CoA + GDP + Pi = succ + GTP + CoASH   Keq ≈ 1

The reaction is readily reversible [32,33], although in the citric acid cycle, net flux is toward the synthesis of succinate and GTP. The mammalian enzyme is typically referred to as being guanine nucleotide specific, but there is also an adenine nucleotide specific enzyme [34] and the spinach enzyme is adenine nucleotide specific [32]. We will focus on the guanine nucleotide-specific reaction, although the arguments are equally applicable to the adenine nucleotide-specific activity. The intramitochondrial guanine nucleotide energy state ([GTP]_f_/[GDP]_f_[Pi]) can be inferred by assuming the reaction is near equilibrium:[GTP]/[GDP][Pi] = ([CoASH]/[succ-CoA])/[succ]   Keq ≈ 4

Hansford and Johnson [35] measured the concentration of coenzyme A and succinyl-CoA in mitochondria isolated from rabbit heart while oxidizing various substrates and at varying rates of oxygen consumption. Table 1 summarized their data for mitochondrial suspensions oxidizing 100 µM palmitoyl-L-carnitine with 0.5 mM malate added as a source of oxaloacetate (their Table IV). ATP (0.5 mM) was added and then the respiratory rate was increased by adding purified hexokinase (the medium contained 10 mM glucose). Although malate was added to provide oxaloacetate to start and maintain the citric acid cycle, during fatty acid oxidation, succinyl-CoA is produced from citrate. The succinate concentration should not exceed that required to maintain cycle activity (0.3 mM was assumed for the calculations in Table 1). As shown in Table 1, the assumption of near equilibrium results in calculated [GTP]_f_/[GDP]_f_[Pi] values consistent with the extramitochondrial [ATP]/[ADP][Pi] for similar experimental conditions.

### 2.7. The Intramitochondrial Energy State Inferred by Metabolic REACTIONS Localized to the Matrix: D. Acetate-CoA Ligase)

Acetate is produced in mammals by a number of reactions, including ethanol oxidation and ketogenesis, and by bacteria in the gut [36,37,38,39]. The concentration in the blood of humans under normal conditions is about 40 µM, increasing to 180 µM after an overnight fast [39,40]. The rate of oxidation rises rapidly with an increase in plasma concentration. In isolated perfused rat hearts, an addition of 2 mM acetate to a perfusion medium containing 5 mM ^14^C-glucose suppressed ^14^CO_2_ production by 50%, replacing 50% of the glucose oxidation with oxidation of unlabeled acetate [36]. Oxidation of acetate occurs in the mitochondrial matrix after activation by acetate-CoA Ligase:Acetate + ATP + HSCo-A = acetyl-CoA + ADP + Pi   Keq ≈ 0.22

The equilibrium constant of 0.22 was determined by Guynn and Veech [41] (38 °C, pH 7.0, ionic strength 0.25; 1 mM Mg^2+^). In order for this reaction to carry out net synthesis of acetyl-CoA, the mass action ratio in vivo must be less than the equilibrium constant.
[acetyl-CoA]/[HSCoA] × 1/[acetate] × [ADP][Pi]/[ATP] < 0.22

The concentration of acetate in the mitochondrial matrix is likely similar to, or less than, that in the blood. Acetate is a weak acid. It does not require a transporter to pass through most membranes and can also be transported by facilitated diffusion on the monocarboxylate transporter. Less is known about the intramitochondrial [acetyl-CoA]/[HSCoA] ratio in vivo. The ratio has been reported, however, for isolated heart mitochondria oxidizing pyruvate, which may be considered metabolically similar to isolated perfused heart oxidizing glucose. In Table 2, the [acetyl-CoA]/[HSCoA] for heart mitochondria oxidizing pyruvate [35,42] has been used to calculate the mass action ratio for acetate-coenzyme A ligase for two hypothetical conditions: (1) a matrix energy state the same as that in the cytoplasm (approximately 5 × 10^4^ M^−1^); and (2) assuming ATP^4-^/ADP^3−^ exchange and a membrane potential of −160 mV, where the matrix [ATP]/[ADP] would be less than that in the cytoplasm by a factor of about 500.

Bethencourt et al. [36] reported that when an isolated rat heart is perfused with 5 mM glucose, adding 2 mM acetate to the perfusion medium results in rapid uptake and oxidation of acetate with the expected proportional suppression of glucose oxidation. This indicates high rates of net synthesis of acetyl-CoA by intramitochondrial acetate-CoA ligase, and therefore, a mass action ratio less that the equilibrium constant. As seen in Table 2, the mass action ratio inferred for an energy state of 5 × 10^4^ M^−1^ is appropriately smaller than the equilibrium constant, whereas for an energy state of 1 × 10^2^ M^−1^, the mass action ratio is substantially greater than the equilibrium constant. This is consistent with the energy state in the mitochondrial matrix of perfused rabbit hearts not being significantly different from that in the cytoplasm.

Interestingly, an addition of 2 mM acetate had no effect on the rate of palmitate oxidation when the rabbit hearts were perfused with a medium containing 0.72 mM palmitate [36]. This is consistent with palmitate being activated by long-chain fatty acid-HSCoA ligase, for which the products are AMP and pyrophosphate (PPi), not ADP and Pi. The PPi is then hydrolyzed (Keq ≈ 3500 [43]), making activation of long chain fatty acids overall highly exergonic and irreversible:Palmitate + ATP + HSCoA = palmitoyl-CoA + AMP + PPi   Keq ≈ 1
PPi = 2 Pi   Keq ≈ 3500

As a result, long chain fatty acids can “out compete” acetate for intramitochondrial coenzyme A. Although initial activation of long chain fatty acids occurs in the cytoplasm, after the carnitine derivatives are transported into the matrix, acylcarnitine transferase resynthesizes acyl-CoA. Hansford and Johnson [35] reported that incubation of isolated heart mitochondria in a medium containing 100 µM palmitoyl-L-carnitine, 0.5 mM L-malate, and 0.1 mM Mg^2+^ resulted in the matrix [acetyl-CoA]/[HSCoA] ratio increasing to >100, high enough to prevent net activation of acetate even at an energy state of 5 × 10^4^ M^−1^. This is consistent with the observation by Bethencourt et al. [36] that although adding 2 mM acetate to the perfusate of isolated rabbit hearts replaced 50% of glucose oxidation, it did not significantly reduce oxidation of palmitate.

## 3. Overview

Evaluation of several energy dependent metabolic reactions occurring in the mitochondrial matrix shows that under physiological conditions, the energy state ([ATP]_f_/[ADP]_f_[Pi]_f_) in the matrix is not different from that in the cytoplasm. This is observed for suspensions of isolated mitochondria and as well as for cells and tissues. A corollary is that, under physiological conditions, exchange of ADP and ATP across the mitochondrial inner membrane is by electroneutral exchange, i.e., the difference in charge between ADP and ATP at physiological pH is compensated by proton co-transport with ATP or some other mechanism that results in the exchange occurring with no net charge. It is important for the eukaryotic metabolism that oxidative phosphorylation and glycolysis be coupled to, and regulated by, a common energy state [44]. The metabolic reactions in the two compartments arise from independent viable organisms and have similar chemistry, and they must work together over a broad range rates of ATP consumption and essential regulatory activities. An example of this coordinated regulation is glucose sensing in the pancreatic beta cell function in diabetes [45,46,47]. Although the primary glucose sensor is cytoplasmic glucokinase, the resulting glycolytic production of pyruvate for oxidation by oxidative phosphorylation is essential to control insulin release. Moreover, in hypoglycemia the deficiency in glucose and thereby of glycolytic production of ATP and pyruvate might be expected to be cytotoxic for β-cells. The increase in ADP, however, activates intramitochondrial glutamate dehydrogenase and thereby amino acid oxidation to maintain cell viability. The glutamate dehydrogenase activation is consistent with activation by an [ADP] similar to [ADP]_f_ in the cytoplasm [45,48], i.e., consistent with [ADP]_f_ in the cytoplasm and matrix being nearly the same.

## Figures and Tables

**Table 1 ijms-23-05550-t001:** Inferring the intramitochondrial [GTP]_f_/[GDP]_f_[Pi] by assuming near equilibrium of the succinate thiokinase reaction.

[CoASH]/[succCoA] *	[ATP]/[ADP] External ^#^	O_2_ rate	[GTP]_f_/[GDP]_f_[Pi] in the Matrix *	[GTP]_f_/[GDP]_f_ in the Matrix *
<0.01	43	0.048	>3.33 × 10^5^ M^−1^	>333
0.011	40	0.064	3.0 × 10^5^ M^−1^	300
0.057	25	0.117	3.3 × 10^4^ M^−1^	33
0.33	2	0.171	1.0 × 10^4^ M^−1^	10

* The data points in columns 1–3 are from Hansford and Johnson [35] for a suspension of rabbit heart mitochondria provided with 100 µM palmitoylcarnitine and 0.5 mM malate with the rate increased by adding purified hexokinase (medium having 5 mM glucose). For calculation of the values in columns 4 and 5, the concentration of intramitochondrial succinate was assumed to be 0.3 mM and inorganic phosphate to be the same as in the suspending medium (10 mM). ^#^ The extramitochondrial [ATP]/[ADP] values reported by the authors are included. At the total concentration of ATP and ADP used (0.5 mM) and Mg^2+^ of 10 mM, however, [ADP] is so low that transport into the mitochondria is limiting. This results in low extramitochondrial [ATP]/[ADP]. The rate of oxygen consumption is in µg atoms O_2_/min/mg protein.

**Table 2 ijms-23-05550-t002:** Inferring the intramitochondrial [ATP]_f_/[ADP]_f_[Pi]_f_ using the acetate-CoA ligase reaction.

	[Acetyl-CoA]/ [HSCoA]	[Acetate]	Mass Action Ratio [ATP]/[ADP][Pi] = 5 × 10^4^	Mass Action Ratio [ATP]/[ADP][Pi] = 1 × 10^2^
Table V *	2	2 × 10^−4^	0.2	100
Table VII *	2.8	2 × 10^−4^	0.28	140
Table V *	2	2 × 10^−3^	0.02	10
Table VII *	2.8	2 × 10^−3^	0.028	14
Figure 8 *	1	2 × 10^−4^	0.1	50

The intramitochondrial [acetylCo-A]/[HSCo-A] ratios for isolated heart mitochondria are taken from * Hansford and Johnson [35] (1975, Tables V and VII) for rabbit heart and from LaNoue et al. [42], their Figure 8 for rat heart. The mitochondrial media used by Hansford and Johnson were contained 10 glucose, 0.5 mM ATP, and 0.1 mM MgCl and either 0.8 mM pyruvate and 0.5 mM L-malate (Table V) or 0.55 mM pyruvate, 1 mM 2-oxoglutarate, and 0.5 mM L-malate (Table VII). LaNoue et al. [42] added 2 mM pyruvate, 1 mM malate, and 5 mM MgCl_2_ to the suspending medium. Bethencourt et al. [36] reported that when isolated rat hearts were perfused with a 5 mM ^14^C-glucose addition of 200 µM acetate to the perfusion medium, it did not significantly suppress glucose oxidation, whereas, adding 2 mM acetate suppressed glucose oxidation by about half. The mass action ratios were calculated for each acetate concentration assuming the matrix energy state (1 mM Mg^+2^) was either 5 × 10^4^ M^−1^ or 1 × 10^2^ M^−1^. The former is the approximately the energy state in cytoplasm of heart tissue, while the latter is a hypothetical matrix energy state assuming electrogenic ATP-ADP exchange and a membrane potential near −160 mV. In order for acetate-CoA ligase to catalyze net synthesis of acetyl-CoA, the mass action ratio must be less than the equilibrium constant.

## Data Availability

Not applicable.

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
