# Peer review of "Integration of Eukaryotic Energy Metabolism: The Intramitochondrial and Cytosolic Energy States ([ATP]f/[ADP]f[Pi])"

_ijms, 2022, doi:10.3390/ijms23105550_

Round 1

Reviewer 1 Report

Wilson and Matchsinsky manuscript reports about the crucial equilibrium in the metabolic energy production, in the form of ATP, and its consumption, between the cytosol and the mitochondrial matrix. They find that ADP in both compartments is selectively bound and the free concentrations are much lower than the total measured concentrations. In addition, they report that under physiological conditions, the transport of ADP and ATP across the mitochondrial inner membrane is "electroneutral".

Page 3 out of 10, lines 7-9: authors report the Kd of ATP and ADP for the ADP/ATP carrier from valid old papers. However, they should know the more recent papers on the recombinant human ADP/ATP carrier reporting about ATP and ADP Km for the ADP/ATP carrier that sounds more specific/precise given that these last estimations were performed on the recombinant protein reconstituted in proteoliposomes. In this reviewer opinion the authors should take in considerations those Km, in their discussion also due to the fact that the Km for ATP appears slightly lower than the Km for ADP in that paper, that should be cited in their manuscript.:

Todisco et al., Biochemical Pharmacology 2016: https://pubmed.ncbi.nlm.nih.gov/26616220/

About the ADP or ATP binding site exposition in this reviewer opinion the authors should delete or reword the sentence at lines 106-109 due to the fact that the ADP or ATP binding regions are further different, although similarly located, beyond the orientation of the carrier in the detergent permeabilized mitochondria. On this concern the authors should read:

Pietropaolo et al., BBA Bioenergetics 2016: https://pubmed.ncbi.nlm.nih.gov/26874054/

However, they should also know that kinetics of the ADP/ATP carrier shows that the affinity of ATP and ADP for the ADP/ATP carrier substrate binding region can be slightly different if the affinity constants are measured in proteoliposomes or in reconstituted mitochondria, not only due to protein orientation: https://www.sciencedirect.com/science/article/pii/S0005273608001442

In this reviewer opinion and on this concern / and more in general in their discussion the authors should briefly report about the fate of the phosphate starting from the paper about phosphate transporters with specific reference to

  • those coded by the human SLC25A23, SLC25A24, SLC25A25, SLC25A31 involved in the translocation of ATP-Mg/Pi (https://www.jbc.org/article/S0021-9258(19)71094-1/fulltext) ;
  • the dicarboxylate carrier coded by SLC25A10 in humans involved in the translocation of organic acids in exchange with phosphate (https://pubmed.ncbi.nlm.nih.gov/29211846/);
  • the phosphate carrier coded by SLC25A3 in humans involved in the symport of phosphate together with H+ (https://pubmed.ncbi.nlm.nih.gov/8718866/)

Above all the last sounds important for the author discussion due to the fact that it was proposed that the rate of phosphate import into mitochondria controls the rate of ATP production by oxidative phosphorylation. For a review about yeast counterpart of the above cited human mitochondrial transporters the authors should read https://pubmed.ncbi.nlm.nih.gov/20533899/.

Page 8 out of 10, lines 358-360: the authors report:

“….under physiological conditions, exchange of ADP and ATP across the mitochondrial inner membrane is by “electroneutral exchange”, i.e. the difference in charge  between ADP and ATP at physiological pH is compensated by proton co-transport with ATP or some other mechanism that results in the exchange occurring with no net charge.”

However, this reviewer retains that there might be an error of definition. On this regard in the above-reported review or in the following Klingenberg review: https://www.sciencedirect.com/science/article/pii/S0005273608001442 the authors will also find indications about modes of transport catalysed by mitochondrial transporters and driving forces.

On this reviewer opinion the authors should provide a clearer explanation about what they mean with electrogenic or electroneutral modes of transport by providing a figure in which they include pH, potentials, protons and the “other mechanisms” they have in mind before this reviewer can continue to evaluate this manuscript.

Conversely, if the authors are stating that the ADP / ATP mode of transport led by the ADP/ATP carrier is not electrogenic as previously proposed in literature, they should clearly state it and discuss in more detail their proposal, by discussing previous literature on the topic.

Reviewer 2 Report

This paper showed very interested phenomenon and analysis in organelle. The authors said that ADP in both the cytosol and matrix is selectively bound and the free concentrations are much lower than the total measured concentrations and the transport of ADP and ATP across the mitochondrial inner membrane is by electroneutral exchange diffusion.  This paper showed the catch of the source of energy in eukaryotes. 

Author Response

The review indicates the reviewer has understood our focus on the thermodynamics and agrees with the importance of the measurements.  No changes are planned.

Thank you

Round 2

Reviewer 1 Report

I thank the authors for their comments. However, I do not agree with them on several points.

Although the paper is not focused on the mechanism of transport of adenine nucleotides and Pi, the authors arrive to a conclusion about electroneutrality or electrogenic exchange which lacks of a more detailed context and might be misleading.

For context I mean that they have to state if the electroneutrality or the electrogenic mechanism refer to the protein or to the compartments.

I appreciated that they removed the misleading sentence from the abstract. However, they should check their sentences in the manuscript still referring to electrogenic/electroneutral exchange. 

The co-transport of a proton with ATP is questionable.

The sentence "some other mechanism that
results in the exchange occurring with no net charge" sounds trivial.

Also with respect to the Km values of ATP and ADP, I do not agree with authors opinion. Indeed, it is well known that if they consider the Km or Kd from purified mitochondria, they are including in those estimations all the other transporters able to translocate ATP across mitochondrial membrane (I.e., the ATP-Mg/Pi carrier coded by SLC25A23, SLC25A24, SLC25A25 and SLC25A41...and other systems that will not be cited here...).

Furthermore, they have to discriminate between protein systems able to bind and modify ADP or ATP (as those cited by the authors, ATPsynthase, carbamoyl phosphate synthase and so on) and systems important for nucleotides translocation.

Thus, in this reviewer opinion, and if the editor agrees the authors can also choose to not discuss the possible carrier systems involved in nucleotide translocation.

However, they should find the way to explain that their analyses do not include all the carrier systems  involved in nucleotide translocation and they should explain why/how they can arrive to their conclusion without consider all the exchange reaction mediated by those carriers.

Surely, any reference to a electroneutrality or electrogenic mechanisms should be avoided, if they cannot provide a more detailed context about what they are estimating.

Author Response

Adenine nucleotide transport from the mitochondrial matrix to the cytoplasm is essential to eukaryotic metabolism due to oxidative phosphorylation being the primary source of ATP for most eukaryotes whereas most ATP consumption is in the cytoplasm. The primary role of that ATP is to provide the energy needed to carry out energy requiring reactions, both chemical and physical, by coupling hydrolysis of the terminal phosphate to the energy consuming reactions. A direct measure of the available energy is the energy state ([ATP]/[ADP][Pi]).  ATP, ADP, and Pi must be transported across the inner mitochondrial membrane, typically with net transport of ATP out of and of ADP and Pi in to the matrix.  Our focus is on determining the energy state in the mitochondrial matrix under in vivo conditions so that it can be compared to that in the cytoplasm, which has been established.  Knowledge of the energy state is essential to understanding energy metabolism in each compartment and as a unified whole. Energy state is a thermodynamic parameter (like equilibrium constants, free energy, etc.) and a state function.  As such, for any given value of the energy state the free energy available from hydrolysis of the terminal phosphate is not dependent on the reaction mechanism(s) involved or rates at which those reaction(s) occurred.  Knowledge of the energy states in the matrix and cytoplasm does, however, place essential constraints on the properties of any proposed mechanisms.  Any mechanism proposed for the transport of ATP, ADP, and Pi between the compartments must predict outcomes (states) consistent with the relationship of the energy states in the two compartments. Because our focus is on determining the energy state, there is minimal discussion of the many proposed mechanisms for adenine nucleotide and Pi transport across the mitochondrial membrane.  Examples are noted only to illustrate the need for considering thermodynamic constraint when discussing proposed mechanisms, constraints that need to be applied to any/all proposed mechanisms.  The key observation is that the energy state in the mitochondrial matrix in vivo is not significantly different from that in the cytoplasm.  This is very important for understanding eukaryotic metabolism in general and essential to any discussion of adenine nucleotide transport. As general examples: 1. if transport is proposed to be rate limiting for any reason, the matrix energy state should be appropriately higher than that in the cytosol to account for the loss in energy (and reversibility) in transport;  2. If transport is proposed to input energy for any reason the energy state in the cytoplasm would be expected to be appropriately higher than that in the matrix.